# Serum-Derived Exosomal MicroRNA Profiles Can Predict Poor Survival Outcomes in Patients with Extranodal Natural Killer/T-Cell Lymphoma

**DOI:** 10.3390/cancers12123548

**Published:** 2020-11-27

**Authors:** Kyung Ju Ryu, Ji Young Lee, Myung Eun Choi, Sang Eun Yoon, Junhun Cho, Young Hyeh Ko, Joon Ho Shim, Won Seog Kim, Chaehwa Park, Seok Jin Kim

**Affiliations:** 1Department of Health Sciences and Technology, Samsung Advanced Institute for Health Sciences and Technology, Sungkyunkwan University, Seoul 06351, Korea; kjryu@skku.edu (K.J.R.); jyoung0319@naver.com (J.Y.L.); skkucme@naver.com (M.E.C.); joonho1224@skku.edu (J.H.S.); wskimsmc@skku.edu (W.S.K.); cpark@skku.edu (C.P.); 2Division of Hematology and Oncology, Department of Medicine, Samsung Medical Center, Sungkyunkwan University School of Medicine, Seoul 06351, Korea; dtd43@daum.net; 3Department of Pathology and Translational Genomics, Samsung Medical Center, Sungkyunkwan University College of Medicine, Seoul 06351, Korea; junhun.cho@samsung.com (J.C.); yhko310@skku.edu (Y.H.K.); 4Samsung Genome Institute, Samsung Medical Center, Sungkyunkwan University School of Medicine, Seoul 06351, Korea

**Keywords:** exosome, NK/T-cell lymphoma, biomarker, microRNA cancer

## Abstract

**Simple Summary:**

Exosomes containing microRNAs (miRNAs) might have utility as biomarkers to predict the risk of treatment failure in extranodal NK/T-cell lymphoma (ENKTL). The aim of our study was to assess the prognostic value of serum-derived exosomal miRNA profiles in patients with ENKTL. The top 20 upregulated miRNAs in patients with poor outcomes and 16 miRNAs upregulated in tumor cell lines identified five candidate miRNAs (miR-320e, miR-4454, miR-222-3p, miR-21-5p, and miR-25-3p). Among these, increased levels of exosomal miR-4454, miR-21-5p, and miR-320e were associated with poor overall survival. These three miRNAs were overexpressed in NKTL cell lines that were resistant to etoposide, and the transfection of NKTL cell lines with miR-21-5p and miR-320e induced an increase in expression of the proinflammatory cytokines. Upregulation of these exosomal miRNAs in treatment-resistant cell lines suggests they have a role as biomarkers for the identification of ENKTL patients at high risk of treatment failure.

**Abstract:**

Exosomes containing microRNAs (miRNAs) might have utility as biomarkers to predict the risk of treatment failure in extranodal NK/T-cell lymphoma (ENKTL) because exosomal cargo miRNAs could reflect tumor aggressiveness. We analyzed the exosomal miRNAs of patients in favorable (*n* = 22) and poor outcome (*n* = 23) groups in a training cohort. Then, using the Nanostring nCounter^®^ microRNA array, we compared them with miRNAs identified in human NK/T lymphoma (NKTL) cell line-derived exosomes to develop exosomal miRNA profiles. We validated the prognostic value of serum exosomal miRNA profiles with an independent cohort (*n* = 85) and analyzed their association with treatment resistance using etoposide-resistant cell lines. A comparison of the top 20 upregulated miRNAs in the training cohort with poor outcomes with 16 miRNAs that were upregulated in both NKTL cell lines, identified five candidate miRNAs (miR-320e, miR-4454, miR-222-3p, miR-21-5p, and miR-25-3p). Among these, increased levels of exosomal miR-4454, miR-21-5p, and miR-320e were associated with poor overall survival in the validation cohort. Increased levels were also found in relapsed patients post-treatment. These three miRNAs were overexpressed in NKTL cell lines that were resistant to etoposide. Furthermore, transfection of NKTL cell lines with miR-21-5p and miR-320e induced an increase in expression of the proinflammatory cytokines such as macrophage inflammatory protein 1 alpha. These studies show that serum levels of exosomal miR-21-5p, miR-320e, and miR-4454 are increased in ENKTL patients with poor prognosis. Upregulation of these exosomal miRNAs in treatment-resistant cell lines suggests they have a role as biomarkers for the identification of ENKTL patients at high risk of treatment failure.

## 1. Introduction

Exosomes are 30–150 nm-sized endosome-derived extracellular vesicles secreted from various types of cells [1,2]. Tumor cells can potentially release exosomes that originate from inward budding of the limiting membrane of multivesicular bodies [3]. Circulating exosomes in the blood and body fluids are involved in cell-to-cell communication and contribute to tumor growth and metastasis because the exosome cargo includes proteins, DNAs, mRNAs, and microRNAs [4,5]. Transfer of exosomal cargo to recipient cells such as adjacent tumor cells and immune cells could play a role in tumor progression. Among the various types of tumor-derived exosomal cargo, exosomal microRNA (miRNA) has been widely studied because deregulation of miRNAs contributes to tumor progression and is common in tumor cells [6,7]. Accordingly, circulating exosomal miRNAs may be potential biomarkers of tumor aggressiveness, as well as having a role in prognosis because of their relative stability compared with free miRNAs in body fluids [8,9].

Extranodal natural killer (NK)/T-cell lymphoma (ENKTL) is an uncommon subtype of non-Hodgkin lymphoma characterized by the predominant involvement of the nasal cavity/nasopharynx and invariable infection of lymphoma cells with the Epstein–Barr virus (EBV) [10]. Although the outcomes of ENKTL have improved with advances in treatment strategies, treatment failure still occurs in a substantial number of ENKTL patients, especially in patients with advanced extra-nasal disease [11,12]. Considering that treatment failure is mainly related to resistance to treatment resulting in early or late relapse, the development of biomarkers predicting relapse after radiologic remission or progression during treatment could help reduce the occurrence of treatment failure and lead to improved outcomes for ENKTL patients. Previous studies have shown miRNA-21 and miRNA-155 could be associated with tumor aggressiveness and poor outcomes of ENKTL, suggesting that targeting miRNA-21 and miRNA-155 might be a viable approach to treating ENKTL [13,14,15]. Given the pro-tumor effect of these miRNAs, exosomal miRNAs may contribute to the aggressiveness of ENKTL. However, the role of exosomal miRNA in ENKTL has never been examined.

For ENKTL prognosis, physicians use the prognostic index for natural killer cell lymphoma (PINK) and PINK-EBV (PINK-E) [16]. However, these indices have limited value in the provision of biological information that may lead to the discovery of therapeutic targets for this disease entity and the treatment outcome could be different within the same risk group. Therefore, we hypothesized that exosomal miRNAs could serve as biomarkers to both predict disease outcomes and identify potential therapeutic targets. In this study, we first analyzed exosomal miRNAs derived from ENKTL patients’ serum according to treatment outcomes in a training cohort and compared these miRNA profiles with those derived from NKTL cell lines. We then validated the clinical relevance of exosomal miRNA profiles using archived serum samples from an independent cohort of ENKTL patients and analyzed their expression in treatment-resistant cell lines.

## 2. Methods

### 2.1. Study Design and Patients

Our primary objective was to develop miRNA profiles to predict prognosis in patients with ENKTL. We used archived serum samples obtained from patients enrolled in prospective cohort studies. Serum samples were stored at −80 °C until analysis and after we had obtained written informed consent. These cohort studies were approved by the Institutional Review Board of Samsung Medical Center. All investigations were conducted according to the principles expressed in the Declaration of Helsinki and its contemporary amendments. Patients in the study population were diagnosed with ENKTL between January 2009 and February 2018 at Samsung Medical Center according to the pathology criteria of the World Health Organization, including positivity for EBV by in situ hybridization [17]. The training cohort (*n* = 45) comprised patients who registered for the first prospective cohort study (2008–2011, NCT#00822731). Patients in the training cohort were grouped according to a favorable (*n* = 22) or poor (*n* = 23) outcome. We measured miRNA read counts for each patient and compared results of the two groups. Increased read counts of miRNAs in patients with a poor prognosis were compared with those in patients with a favorable prognosis in order to identify candidate prognostic markers. To compare serum- and cell-line-derived exosomal miRNA profiles, we also measured the miRNA read counts of ENKTL cell line-derived exosomes.

To validate the miRNA profiles established from the training cohort and ENKTL cell lines, we analyzed the expression of exosomal miRNAs in an independent validation cohort. Patients in the validation cohort were from two prospective cohort studies (2012–2017, NCT#01877109; 2017–ongoing, NCT#03117036) that were performed after the first cohort study. We analyzed the association between each miRNA and survival outcome using validation cohort data and finally developed a list of exosomal miRNAs related to prognosis in ENKTL. We also investigated the correlation between miRNA profiles and the clinical course in the validation cohort using serially collected serum samples at diagnosis, at the interim response evaluation, and at the end of treatment evaluation. We then established miRNA profiles according to the treatment outcomes and analyzed their association with clinical and laboratory parameters including age, performance status, B symptoms, disease stage, serum lactate dehydrogenase (LDH) levels, PINK-E, and EBV DNA titer in blood. To investigate the association of these miRNA profiles with survival outcomes, we collected updated information about survival status in February 2020. The institutional review board at the Samsung Medical Center approved the present study (IRB number 2016-11-040) and all the methods were carried out following the approved guidelines.

### 2.2. Isolation of Exosomes from Serum Samples and Nanostring nCounter Analyses

Serum was differentially centrifuged at 2000× *g* at 4 °C for 10 min and 10,000× *g* at 4 °C for 30 min and then passed through a 0.22 μm filter to remove cell debris. Clarified serum was mixed with ExoQuick (System Biosciences, Palo Alto, CA, USA), incubated at 4 °C for 30 min, and then centrifuged twice at 1500× *g* for 30 min and 5 min, respectively, and the supernatant was discarded. The pellet was resuspended in 200 μL phosphate-buffered saline (PBS). For each sample analyzed, total exosomal RNA was used for miRNA profiling using the Nanostring nCounter^®^ microRNA platform (Version 3) and an nCounter^®^ miRNA custom CodeSet panel (NanoString Technologies, Seattle, WA, USA), as per the manufacturer’s instructions. The read count of each miRNA was subjected to technical normalization by considering the counts obtained for positive control probe sets, followed by biological normalization using ligation controls included in the CodeSet content according to NanoString Technologies’ guidelines. The nCounter results were analyzed using nSolver 4.0 software according to the manufacturer’s instructions. We analyzed 837 human miRNAs in this study. The miRNA list is provided in Appendix A. (nCounter Human v3 miRNA Expression Assay panel, https://www.nanostring.com/products/mirna-assays/mirna-panels).

### 2.3. Cell Lines and Isolation of Exosomes from Cell Culture Medium

The following human NK/T lymphoma (NKTL) cell lines were used: SNK6 (Epstein–Barr virus-positive nasal NK/T-cell lymphoma cell line) cells were cultured in RPMI-1640 medium supplemented with heat-inactivated 10% fetal bovine serum (FBS) and recombinant human IL-2 (PeproTech, Rocky Hill, NJ, USA). Natural killer cell line (NK92MI) cells were cultured in MEM-α medium supplemented with heat-inactivated 20% FBS, penicillin, and streptomycin (Gibco BRL, Grand Island, NY, USA). To isolate exosomes, cells (5 × 10^5^ cells/mL) were suspended in RPMI-1640 medium containing 10% exosome-depleted FBS for 72 h. The cell culture medium was then collected and differentially centrifuged at 300× *g* at 4 °C for 10 min, 2000× *g* at 4 °C for 10 min, and 10,000× *g* at 4 °C for 30 min. After removing cell debris with a 0.22-μm filter (Millipore, Carrigtwohill, Co Cork, Ireland), the cell culture medium was concentrated using an Amicon^®^ Ultra-15 100 kDa device and subsequently concentrated by centrifugation at 3000× *g* at 4 °C using an Allegra^®^ X-15R centrifuge. The final product was mixed with 600 μL ExoQuick-TC (System Biosciences) solution and incubated at 4 °C overnight, then centrifuged twice at 1500× *g* for 30 and 5 min, respectively. After the supernatant was removed, the final pellet was resuspended in 200 μL PBS.

### 2.4. Transmission Electron Microscopy and Nanoparticle Tracking Analysis

For transmission electron microscopy (TEM), exosomes were fixed in 2% paraformaldehyde and transferred onto Formvar-carbon-coated electron microscopy grids. Fixed samples were allowed to absorb for 10–20 min in a dry environment and grids were rinsed in PBS. After removing the supernatant liquid by absorption using filter paper, grids were floated on drops of 2.5% *w*/*v* glutaraldehyde for 5 min. Grids were washed 10 times with distilled water, negative stained with 1% uranyl acetate for 1 min, and then air-dried. Grids were observed using a Hitachi 7700 transmission electron microscope operated at 80 kV.

For nanoparticle tracking analysis (NTA), exosome suspensions with concentrations between 1 × 10^7^/mL and 1 × 10^9^/mL were examined using a Nanosight NS300 (NanoSight Ltd., Amesbury, UK) and the size and quantity of particles were analyzed using NTA software (version 2.3; NanoSight Ltd., Amesbury, UK).

### 2.5. RNA Extraction

Total RNA was isolated from exosome samples using the miRNeasy Serum/Plasma kit according to the manufacturer’s instructions (Qiagen, Valencia, CA, USA). RNA concentration was measured using a NanoDrop ND-100 spectrophotometer (NanoDrop Technologies, Wilmington, DE, USA). The quality of the RNA was analyzed using an Agilent 2100 Bioanalyzer with the total RNA 6000 Nano Chip and Small RNA chip (Agilent Technologies, Santa Clara, CA, USA).

### 2.6. Quantitative Real-Time Polymerase Chain Reaction

For miRNA expression analysis, 10 ng of RNA was mixed with TaqMan MicroRNA Reverse Transcription Kit reagent containing specific miRNA primers and reverse-transcribed according to the manufacturer’s instructions (Thermo Fisher Scientific, Vilnius, Lithuania). Real-Time PCR was performed by diluting the complementary cDNA product in 2x TaqMan Universal Master Mix II (with no AmpErase UNG) and 20x TaqMan microRNA Expression Assay for each mature miRNA to be measured: miR-21-5p (assay ID: 000397), miR-320e (assay ID: 243005_mat), and miR-4454 (assay ID: 461830_mat). Real-time PCR was performed in triplicate for each sample and each miRNA using the ABI PRISM 7900HT platform. Fold changes in miRNA relative to the U6 (assay ID: 001973) and cel-miR-39 (assay ID: 000200) endogenous controls were calculated using the 2^−ΔΔCt^ method.

### 2.7. Western Blot

For Western blot analysis, exosomes were lysed in a RIPA buffer (0.5% sodium deoxycholate, 1% Nonidet P-40, 150 mM NaCl, 50 mM Tris (pH 7.5), 0.1% sodium dodecyl sulfate (SDS), and 1 mM phenylmethylsulfonyl fluoride) and cleared by microcentrifugation (13,000 rpm for 30 min at 4 °C). In total, 30–50 μg of protein samples was electrophoresed on a 4–12% SDS polyacrylamide gel and transferred to nitrocellulose membranes. The membrane blot was incubated with a blocking solution (5% non-fat milk) for 1 h and incubated overnight at 4 °C with primary antibodies including anti-calnexin, anti-Alix, anti-CD63, and anti-CD81 (Santa Cruz Biotechnology, Santa Cruz, CA, USA). The blot was washed with Tris-buffered saline with 0.1% Tween^®^ 20 Detergent (TBST) buffer (50 mM Tris (pH 7.5), 150 mM NaCl, 0.05% Tween 20) and further exposed to horseradish peroxidase-conjugated secondary antibody for 1 h at room temperature. Proteins were visualized using enhanced chemiluminescence reagent (Invitrogen, Carlsbad, CA, USA).

### 2.8. Establishment of Etoposide-Resistant Cell Lines and Cell Viability Assay

To generate drug-resistant cell lines, NK92MI and SNK6 cells were continuously treated with consecutively increasing concentrations of etoposide (Sigma-Aldrich, St Louis, MO, USA) until drug-resistant sublines exhibited at least 2-fold higher IC_50_ (the concentration of a drug is required for 50% inhibition in vitro study) values than the respective parental controls. Established etoposide-resistant cell lines were cultured in a complete medium with etoposide to maintain their drug resistance. Drug effects on cell viability were measured using the Cell Counting Kit-8 reagent (CCK-8) viability assay. Cells (1 × 10^4^) were seeded into 96-well plates and treated with different concentrations of etoposide for 72 h. Then 10 μL of CCK-8 solution (Dojindo Laboratories, Kumamoto, Japan) was added to each well and the plate was incubated for 2 h at 37 °C. Absorbance was read at 450 nm using a microplate reader. Cell viability was expressed as a percentage (optical density of the experimental sample/optical density of control).

### 2.9. miRNA Transfection and Cytokine Array

We used miRCURY LNA miRNAs, miR-21 mimic, miR-320e mimic, and negative control miRNA (scrambled) that were obtained from Exiqon (Copenhagen, Denmark). Cells were transfected with 20 nM miRNA using Lipofectamine™ RNAiMAX (Life Technologies Co., Carlsbad, CA, USA). After 48 h, the conditioned medium was collected from miRNA-transfected cells and cytokine levels were determined using the Proteome Profiler Human XL Cytokine Array (ARY022, R&D Systems, Minneapolis, MN, USA) according to the manufacturer’s instructions.

### 2.10. Transwell Co-Culture Assay

Cells of the human monocyte cell line THP-1 (purchased from ATCC, Rockville, MD, USA) were seeded into a 24-well plate (lower chamber, 3 × 10^5^) and differentiated with 150 nM phorbol 12-myristate 13-acetate (PMA)(Sigma-Aldrich Co., St. Louis, MO, USA) for 24 h, after which the PMA-containing medium was replaced with 10% FBS containing 1 × RPMI medium for 24 h. SNK6 cells transfected with miR-21 and miR-320e were plated in a Transwell chamber (upper chamber pore size, 0.4 μm; Corning Costar, Tewksbury, MA, USA) and incubated for 48 h. Cell densities were chosen in order to have a 1:1 ratio between SNK6 and THP-1 cells co-cultured in 10% exosome-depleted FBS culture in the RPMI medium.

### 2.11. Data Preparation and Statistical Analyses

Data were summarized and normalized using the robust multi-average (RMA) method implemented in Affymetrix^®^ Power Tools (APT). We exported the results of the gene-level RMA analysis and performed differentially expressed gene (DEG) analysis. The statistical significance of the expression data was determined using fold change. For the DEG set, hierarchical cluster analysis was performed using complete linkage and Euclidean distance as a measure of similarity. Gene Enrichment and Functional Annotation analysis of significant probes was performed using Gene Ontology [18] and KEGG (Kyoto Encyclopedia of Genes and Genomes) [19]. All data analyses and visualizations of differentially expressed genes were conducted using R 3.3.3 [20]. Correlations between miRNAs and clinical parameters were analyzed using the chi-square test. Survival outcomes were compared with the log-rank test. Progression-free survival (PFS) was defined as the time from the date of diagnosis to the date of documented disease progression or death. Overall survival (OS) was measured from the date of diagnosis to the date of death due to any cause. All *p*-values < 0.05 were considered significant; two-sided tests were used in all calculations. Statistical analyses were performed with GraphPad Prism 5.0 (GraphPad Software, Inc., San Diego, CA, USA).

## 3. Results

### 3.1. Isolation of Exosomes from Patients’ Serum in the Training Cohort

We isolated exosomes from archived serum samples of ENKTL patients in the training cohort. Serum-derived exosomes ranged in concentration from 0.25 to 14 × 10^12^/mL and had a morphology consistent with that reported previously (Figure 1A,B). The presence of the exosomal markers Alix and CD63 was confirmed by Western blot. Exosomal cargo RNA consisted mainly of small RNAs less than 200 nucleotides (Figure 1C). The training cohort was divided into two groups according to survival outcomes at the median follow-up of 72.8 months (95% CI: 62.4–83.2 months, Figure 1D). In the group with poor outcomes (*n* = 22), all patients died due to early disease relapse or progression after primary treatment and their median PFS and OS were 5.4 months (95% CI: 2.3–8.5 months) and 7.1 months (95% CI: 5.4–8.8 months), respectively. In the group with favorable outcomes (*n* = 23), only six patients experienced relapse. Among these 23 patients, one patient died due to disease relapse and one died from a non-disease-related death. Thus, the OS of the favorable outcome group did not reach the median value (Figure 1E). Consistent with survival outcomes, the characteristics of patients at diagnosis were significantly worse in the group with poor outcomes than in the group with favorable outcomes (Table 1). Thus, all patients in the group with favorable outcomes had Stage I/II ENKTL and a low risk of PINK-E, whereas most patients in the group with poor outcomes had Stage IV ENKTL with a high risk of PINK-E. A comparison of exosomal miRNA expression in these two groups showed differential expression of miRNAs (Figure 1F). The top 20 upregulated miRNAs in the group with poor outcomes were listed according to the order of extent of expression and included 12 miRNAs with two-fold overexpression (*p* < 0.05): miR-320e, miR-4454, miR-4516, miR-630, miR-122-5p, miR-574-5p, miR-22-3p, miR-486-3p, miR-1915-3p, miR-1972, miR-1285, and miR-222-3p (Figure 1G and Appendix A). Although it was not statistically significant, the expression of exosomal miR-21-5p was also increased and is consistent with the previous study reporting increased expression of miR-21 in NKTL cell lines and primary tumor tissue of ENKTL [13]. KEGG analyses of gene expression profiles found these top 20 upregulated miRNAs had common pathways, including cell cycle, cell adhesion, and PI3K and AKT pathways (Appendix A).

### 3.2. NKTL Cell Line-Derived Exosomal miRNA Profiles

Cell-line-derived exosomes were isolated from two NKTL cell lines, SNK6 and NK92MI (Figure 2A). Cell-line-derived exosomes had a diameter of 105.43 ± 9.01 nm and expressed exosomal markers including Alix, CD63, and CD81 (Figure 2B,C). Calnexin, a marker of non-exosome components, was used as a negative control to ensure that there was no detectable cellular contamination. The miRNA region spanning from 10 to 40 nucleotides was represented by a dashed line. NKTL cell-line-derived exosomal miRNAs were enriched in small RNAs compared with cellular miRNAs (Figure 2D). Comparison of miRNA expression showed different patterns in the cell lysates and exosomes (Figure 2E). When we compared the top 20 upregulated miRNAs in both cell lines according to read counts, 16 miRNAs overlapped between SNK6 and NK92MI cells (Figure 2E). Although the read counts of these 16 miRNAs were different between the two cell lines, they all showed increased expression (>2000 counts) (Figure 2E).

### 3.3. Validation of the Prognostic Relevance of Exosomal miRNAs in ENKTL

We compared the top 20 upregulated miRNAs obtained from analysis of the training cohort with the 16 miRNAs that were upregulated in both NKTL cell lines (SNK6 and NK92MI) (Figure 3A). Five miRNAs (miR-320e, miR-4454, miR-222-3p, miR-21-5p, and miR-25-3p) were upregulated in serum-derived and NKTL cell-line-derived exosomes. To validate the prognostic value of miRNA profiles in the training cohort, we analyzed the expression of these exosomal miRNAs in the validation cohort (*n* = 85). Patients were divided into high (above the median value, *n* = 43) and low (below or equal to the median value, *n* = 42) exosomal miRNA groups based on median read counts of each miRNA. Comparison of OS showed that high miR-4454 and high miR-21-5p expression was significantly associated with poor OS in the validation cohort at the median follow-up of 48.2 months (95% CI: 39.6–56.9 months) (Figure 3B,C). Elevated miR-320e expression also tended to be associated with inferior OS, consistent with the results obtained in the training cohort, although this association was not statistically significant (Figure 3D). When we analyzed the associations between these miRNAs and clinical and laboratory characteristics, we found that patients with high miR-21-5p levels had more unfavorable parameters at diagnosis, including increased serum LDH, extranodal involvement, Stage III/IV cancer, blood EBV DNA, and a high risk of PINK-E, when compared with patients with low miR-21-5p levels (Table 2). Patients with elevated miR-4454 expression also showed a tendency to have a greater number of unfavorable parameters than those with low miR-4454 expression (Table 2). Among 85 patients in the validation cohort, we analyzed serial changes in miRNAs using nine patients’ serum samples obtained at the time of diagnosis, at the time of interim analysis, and at the end of treatment. Three patients who experienced a relapse during follow-up showed elevated read counts of miR-4454 and miR-21-5p at the end of treatment, whereas six patients who maintained their remission status showed a decrease in read counts at the end of treatment (Figure 3E). Exosomal miR-320e read counts also increased at the end of treatment in two patients, consistent with the findings for miR-4454 and miR-21-5p (Figure 3E). All patients who showed a decrease in the read counts of these three miRNAs were alive without any evidence of relapse at the end of treatment, whereas the three patients with increased levels of these miRNAs at the end of treatment ultimately died due to disease relapse. To compare the prognostic value of exosomal miRNAs with serum miRNAs, we measured the circulating levels of the three miRNAs in serum and analyzed their association with OS. There was no relationship between serum miR-21-5p and miR-320e with poor OS, whereas serum miR-4454 showed an association with poor OS, similar to the results obtained for exosomal miR-4454 (Appendix A). When patients were grouped according to the levels of expression of these three miRNAs, the OS of patients with increased expression of the three miRNAs was significantly worse than that of patients without it (Figure 3F).

### 3.4. Expression of miR-21-5p, miR-4454, and miR-320e in Treatment-Resistant NKTL Cell Lines

We developed etoposide-resistant cell lines using SNK6 and NK92MI because etoposide is the most commonly used drug in ENKTL patients. Etoposide-resistant NK92MI was established at etoposide concentrations of 50 nM and 100 nM and etoposide-resistant SNK6 was developed at etoposide concentrations of 200 nM and 400 nM. Their cell viability was superior to the control cell at various concentrations of etoposide (Figure 4A,B). These etoposide-resistant NK92MI and SNK6 cells showed a higher expression of miR-21-5p, miR-4454, and miR-320e than the control cells (Figure 4A,B). The amount of RNA was not different between etoposide-resistant cells and control cells and endogenous expression levels of miR-21-5p, miR-320e, and miR-4454 were similar in control cells of NK92MI and SNK6 (Appendix A). The size and concentration of exosomes isolated from etoposide-resistant SNK6 cells were similar to those of control cells (Figure 4C,D). However, the expression of three miRNAs, especially miR-21-5p and miR-320e, was increased in exosomes of etoposide-resistant SNK6 cells (Figure 4E and Appendix A). When we compared the growth curves of cell lines after the knockdown of these miRNAs, we found the growth advantage in etoposide-resistant cell lines was lost and their growth curves returned to the level of control cell lines (Appendix A).

### 3.5. Effect of Upregulated miR-21-5p and miR-320e on Cytokine Production

We transfected NK92MI and SNK6 cells with miR-21-5p and miR-320e for 48 h and analyzed their gene expression profiles to explore the effect of upregulation of miR-21-5p and miR-320e. KEGG analyses of gene expression profiles showed that cytokine–cytokine receptor interactions, PI3K (Phosphoinositide 3-kinases)–Akt (Protein kinase B) signaling, and cell adhesion molecules were targets of miR-21-5p and miR-320e (Figure 5A). After transfection of miR21-5p and miR-320e, the cytokine array showed increased expression of multiple cytokines (Figure 5B). The measurement of pixel density showed increased expression of the macrophage inflammatory protein 1α (MIP-1α) in miR-21-5p transfected SNK6. The transfection of miR-320e induced increased expression of CD40L in NK92MI and MIP-1α, IL5, C-X-C motif chemokine 12 (CXCL12), CD40L, and osteopontin in SNK6 (Figure 5C). As the upregulated proteins downstream of the miRNAs mainly target immune-related pathways, we investigated the effect of exosomal miR-21 or miR-320e on immune cells. SNK6 cells were transfected with miR-21 and miR320e and then co-cultured with the human monocyte THP-1. We found the trend of overexpression of TGF-β, IL-10, CD206, and CCL2 in THP-1 cells, although this was not statistically significant, suggesting that exosomal miR-21-5p and miR-320e from SNK6 cells can induce a transformation into M2-like macrophages (Figure 5D,E). The co-culture of etoposide-resistant SNK6 cells with THP-1 for 48 h also showed that the expression of TGF-β and CCL2 was increased in THP-1, suggesting that exosomes from etoposide-resistant SNK6 cells could induce the transformation into M2-like macrophages (Figure 5F). Furthermore, SNK6 cells showed differential expression of the target genes, especially CD40L and osteopontin, by overexpression and knockdown of miR-320e (Figure 5G).

## 4. Discussion

Tumor cells secret exosomes containing miRNAs into the blood and exosomes could protect miRNAs from degradation in the circulation. Thus, exosomal miRNAs might be more stable than circulating miRNAs and contribute to tumor aggressiveness through cell-to-cell interactions [21]. In this study, comparison of exosomal miRNAs in patients in a training cohort with favorable outcomes with those with poor outcomes allowed us to generate exosomal miRNA profiles predicting a poor prognosis. These serum-derived exosomal miRNA profiles were compared with 16 miRNAs found to be upregulated in two NKTL cell lines, NK92MI and SNK6. Finally, we validated the prognostic value of these upregulated miRNAs in both serum- and cell-line-derived exosomes using serum samples from an independent cohort. We identified the upregulation of exosomal miR-21-5p, miR-320e, and miR-4454 in ENKTL patients with poor survival outcomes in the validation cohort. Serial comparison of pre- and post-treatment expression of these miRNAs also showed that they tended to be correlated with disease relapse in ENKTL patients and were overexpressed in etoposide-resistant cell lines. Our data suggest that exosomal miR-21-5p, miR-320e, and miR-4454 can be used as predictive biomarkers of treatment failure and a poor prognosis in ENKTL patients.

Clinical application of exosomes as biomarkers and therapeutic targets is emerging as an important issue in the management of lymphoma patients because exosomes can be involved in pathogenesis and aggressive tumor cell growth [22]. However, exosome-derived miRNAs have been studied mainly in diffuse large B-cell lymphoma (DLBCL). Increased serum levels of exosomal miR-99a-5p and miR-125b-5p were found to be associated with chemotherapy resistance and shorter PFS in 116 patients with DLBCL, whereas another study reported that exosomal miR-155, miR-let-7g, and miR-let-7i could be used as biomarkers to predict the outcomes of rituximab-containing chemotherapy in DLBCL [23,24]. Similarly, the expression level of an EBV-encoded miRNA was strongly linked to clinical outcomes in elderly patients with EBV-positive DLBCL [25]. These results suggest a role for exosomal miRNAs in regulating disease severity and highlight their potential use as prognostic markers. Nevertheless, the clinical value of exosomal miRNAs in patients with ENKTL has not previously been explored.

We demonstrated that serum-derived exosomal miR-21-5p, miR-320e, and miR-4454 are potential biomarkers in ENKTL patients. MiR-21-5p is known to modulate the expression of multiple cancer-related target genes and is dysregulated in various tumors [26]. Thus, upregulated miR-21-5p could promote tumor growth, metastasis, and treatment resistance. Furthermore, serum exosomal miR-21-5p has been reported to have a clinical impact in patients with esophageal and hepatocellular carcinoma [27,28]. A previous study that analyzed the cellular miRNA profiles of NKTL cell lines, including SNK6, also demonstrated upregulation of miR-21-5p [29]. We found the strong association of elevated exosomal miR-21-5p expression, Stage III/IV ENKTL, and a high risk of PINK-E (Table 2). Among various genetic events related to the pathophysiology of ENKTL, overexpression of the enhancer of zeste homolog 2 (*EZH2*), which is an oncogene, has been reported to influence the aggressiveness of ENKTL [30,31]. A previous study of lung cancer cell lines reported that miR-21-5p could modulate the effect of *EZH2* on the biological behavior of human lung cancer stem cells [32], suggesting that the association of exosomal miR-21-5p with poor prognosis in ENKTL might be due to its effect on *EZH2*. The relation of exosomal miR21-5p to *EZH2* expression in ENKTL needs to be evaluated in future studies. We also demonstrated the increased expression of miR-320e in patients with a poor outcome in the training cohort, although its association with poor OS was not statistically significant in the validation cohort. Although little is known about the role of miR-320e in cancer, increased expression of miR-320e was found to be associated with the occurrence of relapse and poor outcome in patients with Stage III colorectal cancer [33]. In our study of etoposide-resistant NKTL cell lines, miR-320e and miR-21-5p were upregulated in cell lysates and exosomes. In addition, the gene expression profiles of ENKTL cell lines overexpressing miR-21-5p and miR-320e showed activation of pathways related to cytokine–cytokine receptor interactions. Consistent with these results, we found that upregulation of miR-21-5p and miR-320e in NKTL cell lines induced the expression of proinflammatory cytokines including MIP-1α/MIP-1β, CD40L, and CXCL12.

Previous studies showed that inflammatory cytokines and chemokines may be associated with the aggressiveness of ENKTL because hemophagocytic lymphohistiocytosis (HLH), a hyperinflammatory condition that includes the activation of macrophages, is commonly found in advanced-stage ENKTL patients with poor prognosis [34,35]. Furthermore, MIP-1α/MIP-1β were proposed to have an autocrine effect on the metastatic behavior of murine T-cell lymphoma and to be associated with EBV latency III-infected cells [36,37]. Indeed, our previous study demonstrated that MIP-1α increased the growth of patient-derived ENKTL cells and increased serum levels of MIP-1α and was associated with poor outcomes in ENKTL patients [38]. CD40L, also called CD40 ligand or CD154, is a member of the tumor necrosis factor superfamily of molecules and is primarily expressed by activated T-cells. CD40L also plays a role in the stimulation of immune cells, including macrophages. CXCL12 is a chemokine ubiquitously expressed in many tissues and cell types, and the interaction of CXCL12 and its receptor, CXCR4, is involved in multiple processes in the tumor microenvironment [39]. Consistent with the effect of miR-21-5p and miR-320e on the upregulation of proinflammatory proteins, miR-4454 has also been reported to promote the expression of inflammatory markers [40]. These studies are consistent with our finding that exosomal miR-4454 expression was increased in patients with poor survival and in etoposide-resistant ENKTL cell lines. In addition, SNK6 cells transfected with miR-21 and miR320e as well as etoposide-resistant SNK6 cells induced M2 polarization of THP-1 cells. These data suggest that exosomal miR-21 and miR-320e could influence the tumor microenvironment through pro-tumor effects and are consistent with previous studies reporting the role of miR-21 in macrophage M2 polarization in various cancers [41,42,43].

## 5. Conclusions

Taken together, we found that increased serum levels of exosomal miR-21-5p, miR-320e, and miR-4454 could predict poor outcomes in ENKTL patients. Given their upregulation in treatment-resistant cell lines and their ability to increase the expression of proinflammatory cytokines, these three exosomal miRNAs could be used as biomarkers of a high risk of treatment failure in ENKTL patients. Further studies to better understand the mechanisms of action of these exosomal miRNAs are warranted and will aid in the development of risk-adapted treatment strategies for ENKTL.

## Figures and Tables

**Figure 1 cancers-12-03548-f001:**
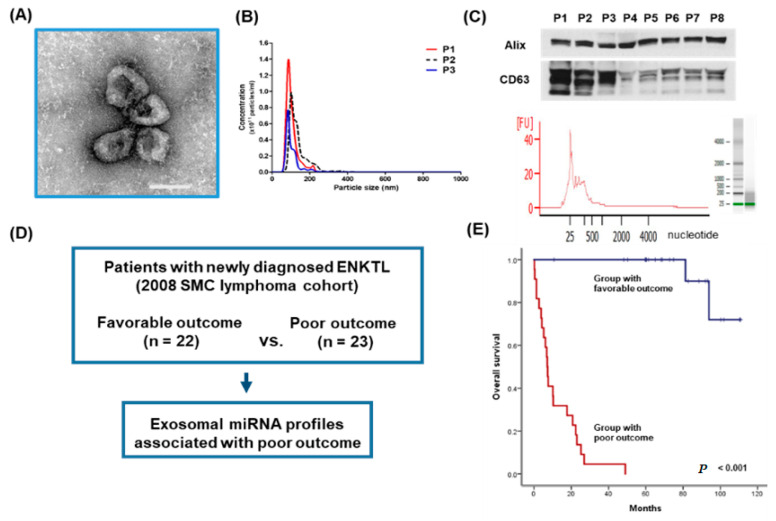
Characterization and miRNA analysis of exosomal miRNA in NK/T-cell lymphoma (NKTL) patients. (**A**) Exosomes were isolated by ExoQuick from equal volumes of serum of NKTL patients; their size and concentration were analyzed by transmission electron microscopy (TEM). Scale bar = 100 nm. (**B**) Nanoparticle tracking analysis. (**C**) Exosomes were analyzed by Western blot for the exosomal protein markers Alix and CD63. RNAs from NKTL serum-derived exosomes were submitted to exosomal miRNA analysis on a total RNA chip using an Agilent 2100 Bioanalyzer. (**D**) Study scheme for the training cohort. (**E**) Kaplan–Meier curves for overall survival of the training cohort. (**F**) Heatmap showing differential exosomal miRNA expression using the Nanostring nCounter^®^ microRNA array between patients with favorable and poor clinical outcomes. (**G**) The top 20 upregulated and downregulated exosomal miRNAs in the group with poor clinical outcomes.

**Figure 2 cancers-12-03548-f002:**
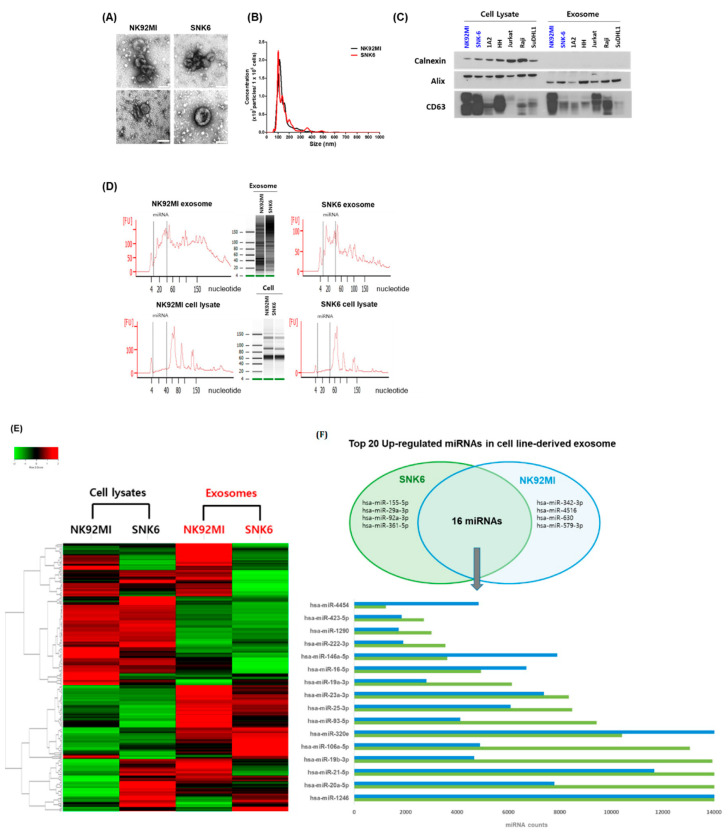
Characterization and miRNA analysis of exosomal and cellular miRNA in NKTL cell lines. (**A**) Exosomes were isolated from lymphoma cell lines by ExoQuick and the size and concentration of the exosomes were assessed by transmission electron microscopy. Scale bar = 100 nm. (**B**) Nanoparticle tracking analysis for NK92MI and SNK6. (**C**) Cell line-derived exosomes were analyzed by Western blot for the exosomal protein markers Alix and CD63. Calnexin, a marker for non-exosome components, was used as a negative control to ensure that there was no detectable cellular contamination. (**D**) NKTL cell-line-derived exosomal and cellular RNAs were measured using the small RNA chip. (**E**) Heatmap showing significantly upregulated and downregulated exosomal and cellular miRNAs. (**F**) The top 20 upregulated miRNAs in exosomes were compared with parent cells; 16 miRNAs were found to overlap between NK92MI and SNK6.

**Figure 3 cancers-12-03548-f003:**
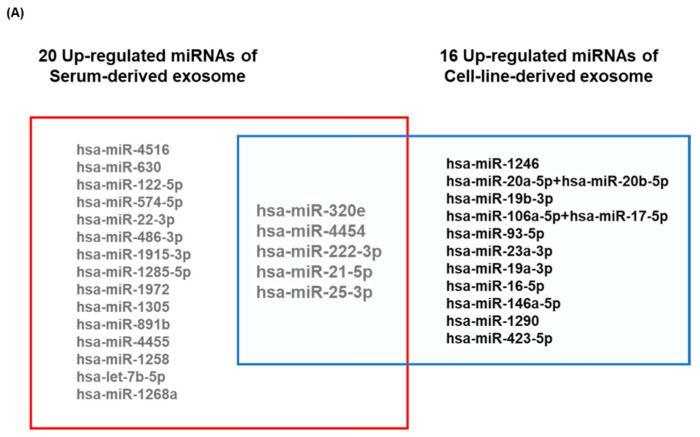
Exosomal miRNA markers correlated with clinical responses. (**A**) Venn diagrams show commonly upregulated miRNAs in both patient-serum-derived and NKTL cell-line-derived exosomes. (**B**) Overall survival of patients with extranodal NK/T-cell lymphoma (ENKTL) according to serum levels of exosomal miR-4454. (**C**) The pretreatment high miR-21-5p group showed poor overall survival compared with the low miR-21-5p group. (**D**) The pretreatment high exosomal miR-320e group showed a tendency of poor survival compared with the low exosomal miR-320e group. (**E**) Serial changes of miR-4454, miR-21-5p, and miR-302e during clinical course. Three relapsed patients showed increased exosomal miR-4454 read counts at the end of treatment (EOT). Patients who relapsed after treatment showed an increase in read counts of miR-21-5p at EOT compared with interim evaluation. Two relapsed patients showed a marked increase in exosomal miR-320e at EOT. (**F**) When the increase in each miRNA was counted as a risk factor for poor prognosis, patients were grouped according to the number of risk factors. Thus, ‘none’ represents the absence of risk factors whereas ‘three’ represents the presence of all risk factors (high miR-21-5p, high miR-320e, and high miR-4454). The overall survival of patients with all three risk factors was significantly worse than that of patients with ‘none’, as well as patients with one or two risk factors.

**Figure 4 cancers-12-03548-f004:**
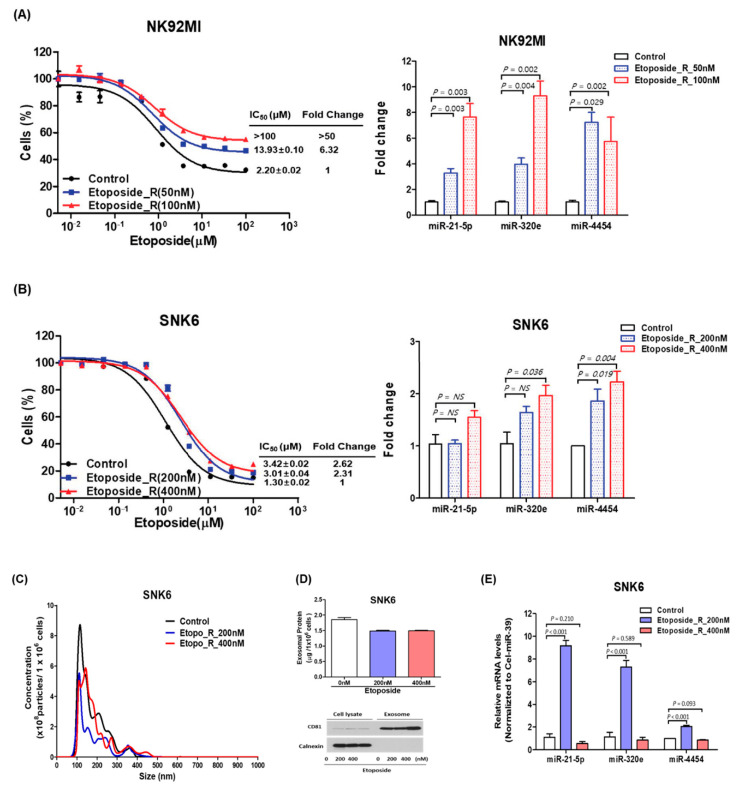
Expression of miR-21-5p, miR-320e, and miR-4454 in etoposide-resistant NKTL cell lines. (**A**) NK92MI (NK92MI_Etopo_R) cells showed significant resistance to etoposide compared with control cells. Cells were incubated with different concentrations of etoposide for 72 h and subjected to the CCK-8 assay. The expression levels of miR-21, miR-320e, and miR-4454 were increased in etoposide-resistant cells. (**B**) SNK6 (SNK6_Etopo_R) cells showed significant resistance to etoposide and increased expression levels of miR-21, miR-320e, and miR-4454 compared with control cells. (**C**) Nanoparticle tracking analysis showed no difference among control SNK6 cells and etoposide-resistant SNK6 cells. (**D**) Exosome amount and marker expression were not different between etoposide-resistant cell lines and control cells. (**E**) The expression levels of miR-21, miR-320e, and miR-4454 in exosomes isolated from SNK6 cells resistant to etoposide 200 nM were higher than in those of control cells.

**Figure 5 cancers-12-03548-f005:**
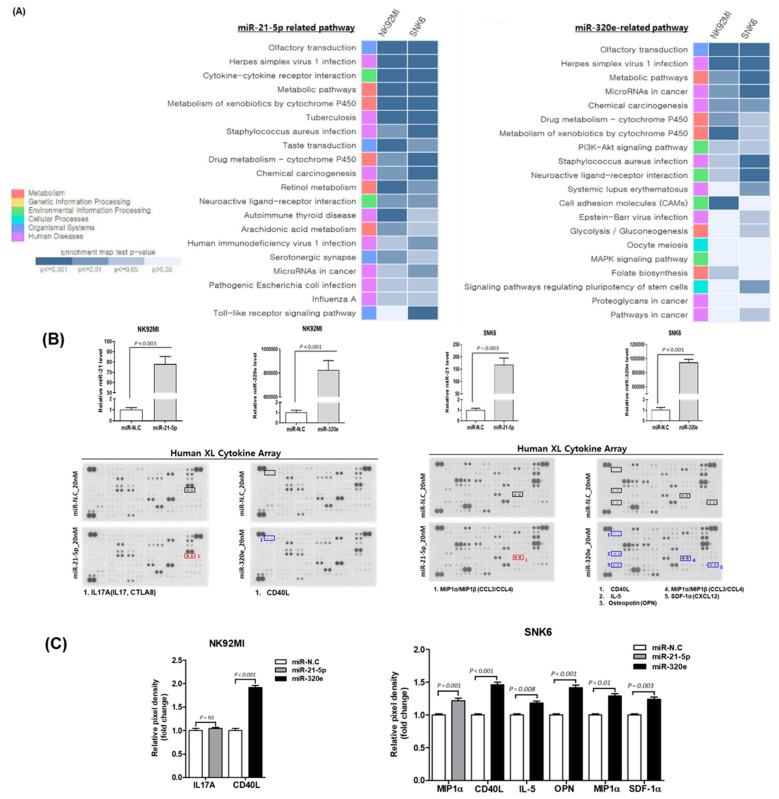
Effect of upregulated miR-21-5p and miR-320e on cytokine production. (**A**) KEGG (Kyoto Encyclopedia of Genes and Genomes) pathway analysis of dysregulated miRNAs was performed for NK92MI and SNK6 cells that were transfected with miR-21-5p mimic and miR-320e mimic. The list of target genes corresponding to up- and downregulated miR-21-5p and miR-320e. (**B**) After transfection of miR-21-5p and miR-320e in NK92MI and SNK6 cells, human cytokine array analysis using a culture supernatant of transfected cell lines was performed. Individual cytokines were spotted in duplicate. Cytokines induced by transfection of cells with miR21-5p are indicated by red squares, while those induced by transfection with miR320e are indicated by blue squares. Positive control spots are located at the corners of the human cytokine array. (**C**) The measurement of pixel densitometry confirmed the increased expression of several cytokines, mainly in miR-320e-transfected SNK6 cells. miR-N.C (negative control); miR-21-5p (miR-21-5p transfection); miR-320e (miR-320e transfection). (**D**) The co-culture of human monocyte THP-1 cells and SNK6 cells transfected with miR-21 showed the trend of overexpression of TGF-β, IL-10, CD206, and CCL2 in THP-1 cells, although this was not statistically significant. (**E**) The co-culture of miR-320e-transfected SNK6 cells with THP-1 cells showed a significant increase in CD206, a marker for M2-like macrophages. (**F**) The co-culture of etoposide-resistant SNK6 cells with THP-1 for 48 h showed a significant increase in TGF-β and CCL2 expression in THP-1 cells. (**G**) SNK6 cells showed differential expression of the target genes, especially CD40L and osteopontin, by overexpression and knockdown of miR-320e.

**Table 1 cancers-12-03548-t001:** Characteristics of patients in the training cohort.

Characteristics	Groups	Poor Outcomes *n* (%)	Favorable Outcomes *n* (%)	*p*
Age	≤60 years	13 (59)	20 (87)	0.047
	>60 years	9 (41)	3 (13)	
Sex	Male	13 (59)	15 (65)	0.763
	Female	9 (41)	8 (35)	
Performance status	ECOG 0/1	15 (68)	23 (100)	0.004
	ECOG ≥ 2	7 (32)	0	
Serum LDH	Normal	2 (9)	17 (74)	<0.001
	Increased	20 (91)	6 (26)	
Stage	I/II	1 (5)	23 (100)	<0.001
	III/IV	21 (95)	0	
Extranodal involvement	Number 0/1	4 (18)	20 (87)	<0.001
	Number ≥ 2	18 (82)	3 (13)	
Bone marrow	Not involved	8 (36)	23 (100)	<0.001
	Involved	14 (64)	0	
Blood EBV DNA	Not detected	0	23 (100)	<0.001
	Detected	22 (100)	0	
PINK-E risk	Low	0	23 (100)	<0.001
	Intermediate	4 (18)	0	
	High	18 (82)	0	
Primary treatment	SMILE	11 (50)	2 (9)	<0.001
	VIDL	5 (23)	0	
	VIPD	3 (14)	0	
	MIDLE	1 (5)	0	
	CCRT followed by VIDL		10 (44)	
	CCRT followed by VIPD		5 (22)	
	CCRT followed by MIDLE		3 (13)	
	Other	1 (5)	3 (13)	
Relapse or progression	Did not occur	0	17 (74)	<0.001
	Occurred	22 (100)	6 (26)	
Survival outcome	Alive	0	21 (91)	<0.001
	Dead	22 (100)	2 (9)	

ECOG: Eastern Cooperative Oncology Group; LDH: lactate dehydrogenase; EBV: Epstein–Barr virus; PINK-E: prognostic index for natural killer cell lymphoma-EBV; SMILE: steroid, methotrexate, ifosfamide, L-asparaginase, and etoposide; VIDL: etoposide, ifosfamide, dexamethasone, and L-asparaginase; VIPD: etoposide, ifosfamide, cisplatin, and dexamethasone: MIDLE: methotrexate, ifosfamide, dexamethasone, L-asparaginase, and etoposide; CCRT: concurrent chemoradiotherapy.

**Table 2 cancers-12-03548-t002:** Association of exosomal miRNAs with the characteristics of the validation cohort.

Characteristics	miR-4454			miR-21-5p			miR-320e		
	Low *n* (%)	High *n* (%)	*p*	Low *n* (%)	High *n* (%)	*p*	Low *n* (%)	High *n* (%)	*p*
Age									
≤60 years	27 (64)	35 (81)	0.091	30 (71)	32 (74)	0.810	29 (69)	33 (77)	0.471
>60 years	15 (36)	8 (19)		12 (29)	11 (26)		13 (31)	10 (23)	
Sex									
Male	32 (76)	27 (63)	0.240	30 (71)	29 (67)	0.815	27 (64)	32 (74)	0.353
Female	10 (24)	16 (37)		12 (29)	14 (33)		15 (36)	11 (26)	
Serum LDH									
Normal	28 (67)	19 (44)	0.050	30 (71)	17 (40)	0.004	25 (60)	22 (51)	0.515
Increased	14 (33)	24 (56)		12 (29)	26 (60)		17 (40)	21 (49)	
Stage									
I/II	29 (69)	21 (49)	0.078	33 (79)	17 (40)	<0.001	28 (67)	22 (51)	0.188
III/IV	13 (31)	22 (51)		9 (21)	26 (60)		14 (33)	21 (49)	
Extranodal involvement									
Number 0/1	26 (62)	22 (51)	0.384	29 (69)	19 (44)	0.029	27 (64)	21 (49)	0.191
Number ≥ 2	16 (38)	21 (49)		13 (31)	24 (56)		15 (36)	22 (51)	
Bone marrow									
Not involved	39 (93)	35 (81)	0.195	39 (93)	35 (81)	0.195	39 (93)	35 (81)	0.195
Involved	3 (7)	8 (19)		3 (7)	8 (19)		3 (7)	8 (19)	
Blood EBV DNA									
Not detected	20 (48)	13 (30)	0.122	24 (57)	9 (21)	0.001	16 (38)	17 (40)	>0.999
Detected	22 (52)	30 (70)		18 (43)	34 (79)		26 (62)	26 (60)	
PINK-E risk									
Low	22 (52)	18 (41)	0.420	26 (62)	14 (33)	0.021	21 (50)	19 (44)	0.146
Intermediate	9 (21)	8 (19)		7 (17)	10 (23)		11 (26)	6 (14)	
High	11 (26)	17 (40)		9 (21)	19 (44)		10 (24)	18 (42)	
Relapse or progression									
Did not occur	25 (60)	16 (37)	0.052	27 (64)	14 (33)	0.005	21 (50)	20 (47)	0.829
Occurred	17 (40)	27 (63)		15 (36)	29 (67)		21 (50)	23 (53)	
Survival									
Alive	30 (71)	22 (51)	0.075	32 (76)	20 (47)	0.007	29 (69)	23 (54)	0.183
Dead	12 (29)	21 (49)		10 (24)	23 (53)		13 (31)	20 (46)	

LDH: lactate dehydrogenase; EBV: Epstein–Barr virus; PINK-E: prognostic index for natural killer cell lymphoma-EBV.

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
