# Peer review of "Serum-Derived Exosomal MicroRNA Profiles Can Predict Poor Survival Outcomes in Patients with Extranodal Natural Killer/T-Cell Lymphoma"

_cancers, 2020, doi:10.3390/cancers12123548_

Round 1

Reviewer 1 Report

In this report, Kyung JR et. al., defined 20 miRNAs from ENKTL patients’ serum-derived exosome with prognostic value. Furthermore, three miRNAs (miR-320e, miR-4454, miR-21-5p) were showed significant enrichment in an independent ENKTL cohort. Mechanistically, ENKTL cell lines transfected with miR-21-5p and miR-320e could upregulate pro-inflammatory factors, such as IL17A, MIP-1α/MIP-1β, CD40L, and CXCL12.

The novelty of this study as follows, 1) established miRNA screening-validation process from patient-derived serum exosome; b) identified three novel miRNAs showed promising prognostic value for ENKTL patients; c) etoposide resistant cell line expressed high levels of miR-320e, miR-4454, miR-21-5p confirmed from clinical data; and d) authors identify miR-320e, miR-4454, miR-21-5p could be used as biomarkers for ENKTL treatment.

The overall writing is clear as described, but the data is not well-presented in the figures. The following concerns should be addressed before further consideration of this manuscript.

Major:

  • Why do we need to use miRNA instead of PINK-E as a biomarker for ENKTL?
  • The authors mentioned that miRNA-21 and miRNA-155 had been validated with good prognostic value in previous studies, how could explain they didn’t show up in this study?
  • Line 317 Figure 4a if the baseline for resistant cell line exosome cargo load more miRNA than the normal condition? So, the miR-320e, miR-4454, miR-21-5p are not specific enriched in resistant cell line exosome.
  • If the author could find another published database to validate those three miRNAs (for example, like GDC or TCGA) which would strengthen this paper.
  • Since the upregulated proteins downstream of miRNAs are mainly targeted immune-related pathways, why not check transfected three miRNAs’ effect on immune cells, such as macrophage, T cells, instead of tumor cell lines.
  • If the pre-screened 20 miRNAs have a common target/pathway as miR-21-5p and miR-320e (cytokine interaction, PI3K, or cell adhesion)?
  • If three miRNAs combine together have better prognostic value than separate?

Minor points:

  • Figure 1 F) could not see it clearly.
  • Figure 4 C) the pattern looks quite different instead of similar (from 100 nm to 300 nm).
  • Figure 4A-B what’s fold changes? Viable cells?
  • Figure 4B significant changes? Should have statistical analyses.
  • Need quantification of cytokine array’s result.

Some figures need more clearly explained (includes but not only, Figure1C [nt]?, Figure 2B,4C 

Author Response

Reviewer #1 (Comments to the Author (Required)):
In this report, Kyung JR et. al., defined 20 miRNAs from ENKTL patients’ serum-derived exosome with prognostic value. Furthermore, three miRNAs (miR-320e, miR-4454, miR-21-5p) were showed significant enrichment in an independent ENKTL cohort. Mechanistically, ENKTL cell lines transfected with miR-21-5p and miR-320e could upregulate pro-inflammatory factors, such as IL17A, MIP-1α/MIP-1β, CD40L, and CXCL12. The novelty of this study as follows, 1) established miRNA screening-validation process from patient-derived serum exosome; b) identified three novel miRNAs showed promising prognostic value for ENKTL patients; c) etoposide resistant cell line expressed high levels of miR-320e, miR-4454, miR-21-5p confirmed from clinical data; and d) authors identify miR-320e, miR-4454, miR-21-5p could be used as biomarkers for ENKTL treatment. The overall writing is clear as described, but the data is not well-presented in the figures. The following concerns should be addressed before further consideration of this manuscript.

Major:

  1. Why do we need to use miRNA instead of PINK-E as a biomarker for ENKTL?

Response: In 2016, I established the PINK and PINK-E model for ENKTL as the first author of ‘A prognostic index for natural killer cell lymphoma after non-anthracycline-based treatment: a multicentre, retrospective analysis (Kim SJ, et al. Lancet Oncol 2016, 17, 389-400). The PINK-E is a prognostic model mainly based on clinical parameters. So, it can be easily used in clinical practice. However, the treatment outcome could be different within the same risk group of PINK-E because their biological characteristics could be different. Thus, we have to explore additional novel biomarkers for precision medicine in patients with ENKTL. Furthermore, there is no targeted agent specialized for the treatment of ENKTL. The data about miRNAs related with tumor aggressiveness could provide important information leading to the discovery of novel drugs for this disease entity. Therefore, we need to develop genomics-based biomarkers including exosomal miRNAs for ENKTL. I mentioned this part in the discussion (Page 5, line 79 - 82).

  1. The authors mentioned that miRNA-21 and miRNA-155 had been validated with good prognostic value in previous studies, how could explain they didn’t show up in this study?

Response: As we stated in the introduction, miRNA-21 and miRNA-155 had been validated with poor prognostic value in previous studies. Yamanaka, Y et al showed overexpression of miRNA-21 and miRNA-155 was found in NK-cell lymphoma cell lines and primary tumor tissue compared to normal NK cells (Blood 2009, 114, 3265-3275). They suggested targeting miRNA-21 and miRNA-155 might be a useful approach to treating NK-cell lymphoma/leukemia because reducing expression of miR-21 or miR-155 led to up-regulation of PTEN and programmed cell death 4. Likewise, Zhang X et al reported that miRNA-155 expression was higher in patients with stable or progressive disease than in those with partial or complete remission (Oncotarget 2016 Aug 16; 7(33):53808-53819). In our study, miRNA-155-5p was also up-regulated in SNK6, and miRNA-21-5p was increased in patients with poor prognosis (Figure 2 and 3). MicroRNA-155-5p and microRNA-21-5p are forms of mature miRNA-155 and miRNA-21. Thus, our study also has shown the upregulation of those microRNAs that were previously reported in ENKTL, however, our study focused on three microRNAs including microRNA-21-5p. We mentioned them in introduction and results (Page 4, line 74 – 78; Page 14, line 270 - 274).

  1. Line 317 Figure 4a if the baseline for resistant cell line exosome cargo load more miRNA than the normal condition? So, the miR-320e, miR-4454, miR-21-5p are not specific enriched in resistant cell line exosome.

Response: We compared the amount of RNA between etoposide-resistant and control cells, and there was no significant difference of total RNA as well as exosomal RNA among them (Supplementary figure 4). We described these findings in results (Page 16, line 333 – 339).

  1. If the author could find another published database to validate those three miRNAs (for example, like GDC or TCGA) which would strengthen this paper.

Response: As there are few data about those three miRNAs in ENKTL, we could not find them from other database including GDC and TCGA.

  1. Since the upregulated proteins downstream of miRNAs are mainly targeted immune-related pathways, why not check transfected three miRNAs’ effect on immune cells, such as macrophage, T cells, instead of tumor cell lines.

Response: According to your comments, we performed additional experiments to investigate the effect of exosomal miR-21 or miR-320e on immune cells. Thus, after SNK6 cells were transfected with miR-21 and miR320e, they were cocultured with the human monocyte/M0 macrophage cell line, THP-1. We found increased expression of TGF-β, IL-10, CD206 and CCL2 in THP-1 cells. These results suggest that exosomal miR-21 or miR-320e from SNK6 cells could induce the polarization into M2-like macrophages leading to pro-tumor effects. In addition, we performed the coculture of etoposide-resistant SNK6 cells with THP-1. After the coculture of 48 hours, the expression of TGF-β and CCL2 was increased in THP-1. These results suggest that exosomes from etoposide-resistant SNK6 cells could induce the polarization into M2-like macrophages with pro-tumor effects (Page 17, line 361 – 370, Figure 5D-F). We added the methods for these additional experiments (Page 11, line 220 – 227) and described them in discussion (Page 21, line 553 – 557).

  1. If the pre-screened 20 miRNAs have a common target/pathway as miR-21-5p and miR-320e (cytokine interaction, PI3K, or cell adhesion)?

Response: Using DIANA mirPath v.3 web server and TarBase 7.0 data, we performed pathway enrichment analysis of pre-screened 20 miRNAs, which have passed statistical significance thresholds, including 10 miRNAs. We identified 52 enriched KEGG pathways (p < 0.05). Among them, we added the following pathways as supplementary data (Page 14, line 272 – 274, Supplementary figure 2).

  1. If three miRNAs combine together have better prognostic value than separate?

Response: We analyzed the overall survival using the combination of three microRNAs. We added this part in the results (Page 16, line 323 – 325, figure 3F).

Minor points:

  1. Figure 1 F) could not see it clearly.

Response: We increased the size of figure 1F.

  1. Figure 4 C) the pattern looks quite different instead of similar (from 100 nm to 300 nm).

Response: When we compared the diameters of exosomes between etoposide-resistant cells and control cells, there was no significant difference: control SNK6 (114.5 ± 4.8), etoposide 200nM-resistant SNK6 (108.1 ± 2.1) and etoposide 400nM-resistant SNK6 (120.9 ± 6.3) nm.

  1. Figure 4A-B what’s fold changes? Viable cells?

Response: They represent the fold changes of viable cells.

  1. Figure 4B significant changes? Should have statistical analyses.

Response: We added P-values for each bars.

  1. Need quantification of cytokine array’s result.

Response: The pixel densities of proteins in cytokine array were quantified with ImageJ software. When we quantified the expression of cytokines, the difference of IL17A was not significant. Thus, we revised results and the figure 4G and H (Page 17, line 356 – 361).

  1. Some figures need more clearly explained (includes but not only, Figure1C [nt]?, Figure 2B,4C
    Response: In figure 1C and figure 2D, [nt] means nucleotide: Fragments of sizes 25, 500, 2000 and 2000 nucleotides. Thus, we revised figure 1C and 2D. Figure 2B and 4C showed the range of exosomal size (nm).

Reviewer 2 Report

The authors characterise exosomal microRNA profiles and relate these with survival outcomes in patients with natural killer/T-cell lymphoma. The results are very well presented and clear. The authors also transfect the micro RNAs into NK tumor cell lines to provide some mechanistic data and proof on concept for activation of some of the inflammtory pathways they have identified.

Author Response

Reviewer #2 (Comments to the Author (Required)):
The authors characterize exosomal microRNA profiles and relate these with survival outcomes in patients with natural killer/T-cell lymphoma. The results are very well presented and clear. The authors also transfect the micro RNAs into NK tumor cell lines to provide some mechanistic data and proof on concept for activation of some of the inflammatory pathways they have identified.
Response: Thank you so much for your very positive feedback.

Reviewer 3 Report

In this manuscript, the authors reported that three exosomal miR-21-5p, miR-320e and miR-4454 were associated poor survival outcome in extranodal NK/T-cell lymphoma. In addition, Overexpressoin of miR-21-5p and miR-320e induced macrophage inflammatory protein 1α (MIP-1α). Here are some suggestions that the authors should consider in order to improve the paper.

Major comments:

  1. The primer sequences of miRNAs should be shown.
  2. The endogenous expression levels of those miRNAs in cell lines should be shown.
  3. Did the author analyze combined effect of miRNAs in survival outcomes?
  4. In Figure 4A and 4B, the drug resistance phenotype should be re-assayed in knockdown miRNAs cell lines.
  5. In Figure 4G and 4H, the validation should be assayed in overexpression and knockdown miRNAs cell lines. In addition, the effects of MIP-1α on drug resistance should be assayed.
  6. There are several typing, defines and grammar errors throughout the manuscript, which should be reviewed by an English native speaker before submission.

Author Response

Reviewer #3 (Comments to the Author (Required)):
In this manuscript, the authors reported that three exosomal miR-21-5p, miR-320e and miR-4454 were associated poor survival outcome in extranodal NK/T-cell lymphoma. In addition, overexpression of miR-21-5p and miR-320e induced macrophage inflammatory protein 1α (MIP-1α). Here are some suggestions that the authors should consider in order to improve the paper.

Major comments:

  1. The primer sequences of miRNAs should be shown.

Response: We used taqman probe assay that were commercially available. Because the company does not disclose the sequences in case of commercially available taqman probes, we added the ID numbers of taqman probe assay (Page 9, line 179 – 184).

  1. The endogenous expression levels of those miRNAs in cell lines should be shown.

Response: We measured endogenous expression levels of miR-21-5p, miR-320e and miR-4454 in NK92MI and SNK6 cells using qRT-PCR (Page 16, line 331 – 336, Supplementary Figure 4).

  1. Did the author analyze combined effect of miRNAs in survival outcomes?

Response: We analyzed the overall survival using the combination of three microRNAs. We added this part in the results (Page 16, line 323 – 325, figure 3F).

  1. In Figure 4A and 4B, the drug resistance phenotype should be re-assayed in knockdown miRNAs cell lines.

Response: We demonstrated the up-regulation of miRNAs in etoposide-resistant SNK6 and NK92MI in figure 4A and 4B. According to your comments, we compared the growth curves of cell lines after the knockdown of those miRNAs. We found the loss of growth advantage in etoposide-resistant cell lines, thus, their growth curves returned to the level of control cell lines (Page 16, line 339 – 342, Supplementary Figure 5).

  1. In Figure 4G and 4H, the validation should be assayed in overexpression and knockdown miRNAs cell lines. In addition, the effects of MIP-1α on drug resistance should be assayed.

Response: SNK6 cells were transfected with miR-21-5p and miR-320e for 48 hours, respectively, and the expression level of regulatory genes of miR-21-5p and miR-320e were analyzed by qRT-PCR. We found differential expression of target genes, especially CD40L and osteopontin in SNK6 cells by overexpression and knockdown of miR-320e (Page 17, line 370 – 372, Figure 5F). In addition, we analyzed the effects of MIP-1α on drug resistance in SNK6 cells. In our previous study, we demonstrated the inferior outcome of ENKTL patients with increased level of serum MIP-1α and the MIP-1α increased proliferation of primary tumor cells derived from ENKTL patients (Kim, H.S.; Ryu, K.J.; Ko, Y.H.; Kim, H.J.; Kim, S.H.; Kim, W.S.; Kim, S.J. Macrophage inflammatory protein 1 alpha (MIP-1alpha) may be associated with poor outcome in patients with extranodal NK/T-cell lymphoma. Hematol Oncol 2017, 35, 310-316). However, there was no significant difference in the absence or presence of different concentration of MIP-1α on drug resistance in SNK6 cells. Thus, we think the increased expression of MIP-1α was the downstream effect of etoposide-resistant cell lines rather than the cause of etoposide resistance. As our study was not focused on MIP-1α, we did not add these findings. However, we will do further studies regarding them in our next project.  

  1. There are several typing, defines and grammar errors throughout the manuscript, which should be reviewed by an English native speaker before submission.

Response: We transferred our revised manuscript to the English proofreading service. Then, we re-submit this revised version.

Round 2

Reviewer 1 Report

I act ed as a reviewer for the original submission to the Cancers. I raised a few concerns that It thought should be addressed in a revision. The authors have now performed an acceptable revision and have addressed the concerns that were raised during the first round of review. As such, it is my view that the revised manuscript is improved and that this paper is now able for publication in Cancers.

Author Response

I act ed as a reviewer for the original submission to the Cancers. I raised a few concerns that It thought should be addressed in a revision. The authors have now performed an acceptable revision and have addressed the concerns that were raised during the first round of review. As such, it is my view that the revised manuscript is improved and that this paper is now able for publication in Cancers.

Response: I thank you so much for your comments on the revised version.

Reviewer 3 Report

The authors have adequately responded to most of my previous concerns.

Author Response

The authors have adequately responded to most of my previous concerns.

Response: I thank you so much for your comments on the revised manuscript.